# The mediation effects of COVID-19-related traumatic stress symptoms and mentalization on the relationship between perceived stress and psychological well-being in healthcare workers transitioning to a post-pandemic world

**Qian Wang**[1‡*], **You Zhou**[2‡], **Gang Wang**[1,3], **Xinyu Pan**[4], **Sha Sha**[1], **Zhe Wang**[1], **Yinqi Liu**[1], **Tengfei Tian**[1], **Sixiang Liang**[1]

**1** The National Clinical Research Center for Mental Disorders & Beijing Key Laboratory of Mental Disorders, Beijing Anding Hospital, Capital Medical University, Beijing, China, **2** New York Psychoanalytic Society & Institute, New York, New York, United States of America, **3** Advanced Innovation Center for Human Brain Protection, Capital Medical University, Beijing, China, **4** The University of Hong Kong, Pok Fu Lam, Hong Kong

‡ QW and YZ are joint first authors on this work.
* wangqian1313246@163.com

## Abstract

### Background

In context of COVID-19 as a collective trauma and the intense involvement of healthcare workers (HCWs) in the pandemic, perceived stress continues to have a tremendous impact on their psychological well-being. However, few studies have attempted to delineate the underlying mechanisms. This study examined whether COVID-19-related traumatic stress symptoms and mentalization act as mediators.

### Methods

A sample of HCWs (N = 2610) from 22 hospitals in Beijing, China participated in this cross-sectional investigation. Data on their perceived stress, psychological well-being, the impact of event, and reflective function during the COVID-19 pandemic were collected using self-report questionnaires. Different mediating models were tested.

### Results

COVID-19-related stress symptoms and mentalization independently mediate the association between perceived stress and psychological well-being. These two mediators also compose a serial mediation model. In particular, higher perceived stress inhibits the psychological well-being of HCWs through increased severity of traumatic stress symptoms, which in turn is associated with hypomentalizing.

**Data Availability Statement:** All relevant data are within the paper and its Supporting information Files.

**Funding:** The author(s) received no specific funding for this work.

**Competing interests:** No authors have competing interests.

## Conclusion

These findings shed light on the mechanisms underlying the relationship between perceived stress and psychological well-being in HCWs. We strongly recommend incorporating a mentalization framework with trauma-informed practice in prevention and intervention work with this population during this and future healthcare crisis.

## Introduction

The outbreak of COVID-19, the subsequent lockdown measures and the constantly changing landscape of the COVID-19 pandemic have placed unprecedented strains on healthcare systems worldwide. Healthcare workers (HCWs) including nurses, physicians, medical administration staff, and others work under significant pressure and with great responsibilities. It is critical to identify factors that can affect the immediate and long-term psychological well-being of the HCWs in context so that effective preventions and interventions during and beyond the current crisis can be provided [1].

Previous studies have so far consistently demonstrated that the pandemic has a detrimental effect on the mental health and well-being of healthcare workers (HCWs) [2–6]. A range of important factors have been identified either directly or indirectly impact on HCWs' psychological well-being including their work environment, work-related stress, individual's coping mechanism or strategies, and social and organizational support. For instance in work environment, HCWs have been coping with challenges such as more job demands, staff shortage [7], lack of protective equipment, and medical supplies [8]. There was also work-related stress [9, 10] such as concerns about being infected, fear of exposure to themselves and their families [11, 12], risk of death of patients [13] and social exclusion [14]. The psychological well-being of HCWs during COVID-19 can also vary depending on individuals' coping mechanism or strategies [15, 16], social support [9, 17–19] and organizational support [9, 20–22]. This paper aims to broaden the exiting understanding of the psychological well-being of HCWs in crisis examining the relationships among perceived stress, trauma and mentalization as well as their impact on the psychological well-being.

### Critical incident involvement of HCWs in a collective trauma

In March 2020, the World Health Organization has declared the COVID-19 coronavirus outbreak as a pandemic [23]. The global epidemiological and psychological crisis caused by COVID-19 can be experienced by the general population and HCWs as a traumatic event or"collective trauma"[24]. In the healthcare setting, COVID-19-related incidents have also been referred to as critical incidents [25, 26]. A critical incident is defined as"a self-defined traumatic event that causes individuals to experience such strong emotional responses that usual coping mechanisms are ineffective."[27]. It has been found that critical incident involvement is particularly problematic in the healthcare setting, as exposure to a traumatic event including response to COVID-19 could exacerbate the intense job demands especially for those who are intensely involved [28]. The HCWs on the frontline must deal with significant mortality and morbidity among patients caused by COVID-19 as well as their own vulnerability and situations in which patients cannot be helped due to external circumstances. Even for HCWs working on second line, in some cases, witnessing a traumatic event, learning about the event was found to be similar to experiencing the event first-hand [29].

Studies have found that exposure to collective trauma can be linked with traumatic stress and symptoms consistent with posttraumatic stress disorder (PTSD) [30, 31]. According to the meta-analysis conducted by Cooke and colleagues, the pooled prevalence among 14 studies indicated 23.88% of PTSD in the general population [32]. In comparison, HCWs were found to have an even higher risk for PTSD than the general population [33]. Systematic review and meta-analyses of the prevalence of PTSD in HCWs during the COVID-19 consistently reported to be in the range of 10.5%–40% [34, 35].

## Perceived stress and psychological well-being in HCWs during the pandemic

The theoretical definition of psychological well-being and its varying conceptual models can often be traced back to the two distinctive philosophical stances of hedonism and eudaimonism [36, 37]. Hedonic perspective emphasizes on well-being on a subjective level as reflected by studies on reported happiness, life satisfaction and positive affect [38–40]; whereas eudaimonisc approach places more emphasis on the importance of meaning-making, self-realizing and fulfilment in life [36, 41, 42]. Following the eudaimonisc perspective, Ryff and colleagues have developed and have been validating the conceptual foundation and empirical indicators for well-being [42]. Their model proposes six dimensions of psychological well-being, namely, autonomy, personal growth, positive relationships, environmental mastery, purpose in life and self-acceptance [41, 43]. This model and its constructs have been extensively validated and supported by a large quantity of independent, empirical studies across disciplines and in various contexts [42].

Good psychological well-being was found to significantly improve the performance of doctors [44], whereas HCWs who lack psychological well-being may exhibit poor mental and emotional functioning, which in turn affects their ability to enjoy life and their work capacity with patients [44]. This is particularly the case in the current context of COVID-19 as numerous research findings have highlighted the elevated psychological distress in HCWs including burnout, and increased risk of mental illness and suicide during the pandemic [45–50]. These studies have highlighted the immediate psychological distress in HCWs as well as recognized that the COVID-19 pandemic may have long-lasting psychological effects as been after the SARS epidemic [30].

One of the main factors that often associates with psychological well-being is perceived stress. Substantial evidence for the relationship between perceived stress and various constructs of psychological well-being existed before the pandemic [51–53]. In the current context, more than ever, perceived stress continues to affect the psychological well-being of HCWs [54, 55]. It should be noted, although there has been abundant evidence of the psychological effects of working in a healthcare setting during the COVID-19 pandemic, such as depression, anxiety or insomnia [33–35, 49], little work has been done on the psychological well-being focusing on the positive functioning of HCWs as well as the mechanism involved in the relationship between perceived stress and psychological well-being [56].

**The role of mentalization.**   Mentalization refers to the capacity to reflect on oneself or others in thoughts, emotions, motivations, beliefs, or other mental states [57–59]. Reflective function (RF) is another term often used interchangeably with mentalization in the literature and was initially used in the work with parent-infant attachment [60] and borderline personality disorder [61, 62] to operationalize mentalization. Currently, mentalization can be measured directly using an interview-based RF scale [63, 64] or self-report questionnaires [65, 66].

Mentalization has been found strongly related to resilience [65]. It is a valuable psychological resource to promote psychological well-being [67], and it can act as an effective protective

factor for a range of psychopathologies [59, 68], especially for individuals exposed to trauma [59, 69]. Mentalization is also closely related to emotion regulation [67], secure attachment [70], social learning [71], and other psychological functions [72, 73]. Previous studies have found that impairment in mentalization can be associated with psychopathology, such as depression, anxiety, eating disorder, substance abuse, and personality disorder [59, 74, 75]. Specifically, the inability to consider complex models of one's own mind and/or of others (i.e., hypomentalizing) or over-generalization of mentalistic representations of actions without appropriate evidence to support them (i.e., hypermentalizing) has shown to increase the vulnerability of psychopathology [65, 71]. Hypomentalizing in particular has been found to be associated with emotional exhaustion and depersonalization in HCW and negatively related personal accomplishment [76].

In theory, the development of mentalization is significantly influenced by the trauma and conflict within premature attachment relationships [77]. Research evidence supports that early trauma could disrupt the development of mentalization and result in failure of mentalization in adulthood [63]. However, the nature of mentalizing ability is considered dynamic and it can be influenced by a range of environmental factors including stress and trauma. It was found the quality of mentalization depends on stress and arousal [78]. As stress and arousal increase, mentalizing ability decreases [79]. Bateman and Fonagy proposed that mentalizing deficits become more pronounced when arousal level rises. In stressful situations, individuals may regress to "pre-mentalizing mode", losing the ability to observe from the perspective of others and dwelling on emotions that are unrelated to objective reality [61]. This is also observed in clinical practice that many patients who have previously shown stable and high mentalizing capacity appear to have lost the mentalizing ability in the COVID-19 context and struggled to regain mentalizing [80]. In addition, Mayes suggested that the narrow threshold for fluctuation in mentalization is connected to the long history of excessive arousal. People impacted by trauma are more likely to experience severe arousal, which causes a loss of mentalization consequently [81]. This is also confirmed by Fowler who suggested that traumatic events could decrease one's mentalization [82].

## Theoretical framework, aims and hypotheses

Perceived stress refers to the degree to which life situations are appraised as stressful [83] and it can predict post stress-exposure psychopathology [84]. During the pandemic, while HCWs maybe used to manage traumatic events as routine part of their career, the stress they perceive, and the intensity of their responses are likely to be increased when critical incidents occur in an environment where there are chronic or long-term stressors present. Previous findings have indicated that genuine mentalization helps overcome stress [67, 85] and reduces the degree of burnout [76, 86]. Specifically in the study conducted by Safiya and colleague, hypomentalizing was found as a significant positive predictor of burnout dimensions of emotional exhaustion and depersonalization in HCWs during the Pandemic [76].

Limited evidence of mentalizing as a mediator had been found between stress and well-being such as between stress and coping [67], stress and hopelessness [87] and stress and emotional regulation [82, 87]. It is plausible that under certain levels of stress, including general perceived stress and specific COVID-19-related traumatic stress, the mentalizing ability of the HCWs can be negatively affected or temporarily lost, leading to more rigid, non-reflective, or excessive mentalizing about the self and others that can subsequently negatively affect their psychological well-being. Previous research has confirmed that interventions for mentalizing abilities can successfully enhance the mentalization of individuals, which in turn enhance their psychological functioning [59, 88]. In the context of COVID-19 experienced as a global

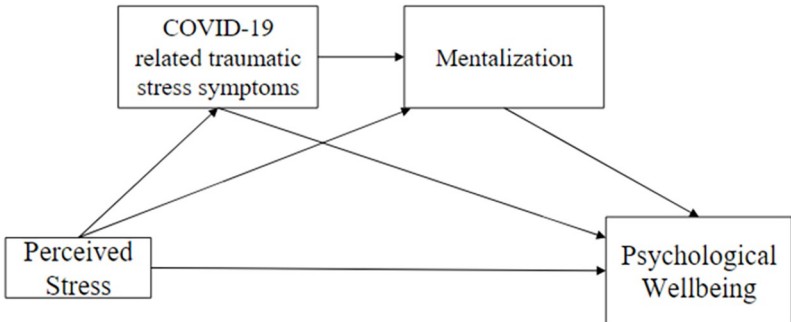

**Fig 1. The hypothesized serial mediation with COVID-19-related traumatic stress symptoms and mentalization as mediators to link between perceived stress and psychological well-being.**

collective trauma and the intense involvement of HCWs in critical incidents in the present and future, this study aims to examine the role of mentalization played in the relationship between stress and psychological well-being. It provides valuable insights into the well-being of HCWs in the later phase of the pandemic as well as information for more effective prevention and intervention strategies that can continue to support HCWs in dealing with the long-term impact of the pandemic and its aftermath.

Taken together, we propose a chained mediation model of traumatic stress symptoms and mentalization to understand perceived stress and psychological well-being in the context of the COVID-19 pandemic (Fig 1). Based on theory and past findings we hypothesized that perceived stress would be negatively associated with the psychological well-being of the HCWs in China during the COVID-19 pandemic (Hypothesis 1). The impact of an event, as an indication of traumatic stress symptoms related to the COVID-19 pandemic can mediate this relationship between perceived stress and psychological well-being in HCWs (Hypothesis 2). The reflective function, the operationalized reference to the capacity of mentalization, can also play a role as a mediator linking perceived stress and psychological well-being (Hypothesis 3). The traumatic stress symptoms and mentalization can act as chained mediators in explaining the relationship between perceived stress and psychological well-being (Hypothesis 4).

## Materials and methods

### Participants

The participant sample was part of a longitudinal project of tracking the impact of COVID-19 on HCWs in Beijing, China. The survey used in this study was a part of the baseline data collection. The sample consisted of 2610 participants from 22 hospitals who had experienced major COVID-19 outbreak incidents in China and recruited from October 2020 to November 2020. The mean age of sample was 36.56 (SD = 8.22) with the majority female HCWs (76.8%). The sample included 523 doctors (20.04%), 1448 nurses (55.48%), 419 medical technicians (16.05%), and 220 administration staff (8.43%). Seventy-eight percent of the participants had educational attainment equivalent to Bachelor's degree or higher. Participating HCWs had varying levels of risk exposure to COVID-19 working across departments such as fever clinic, respiratory department, intensive care unit, emergency department and COVID-19 wards. Approximately 57.7% of them worked directly with COVID-19 positive or exposed patients. More detailed demographic information of the sample was summarized in Table 1.

**Table 1. Characteristics of participants (N = 2610).**

|  | N | % |
|---|---|---|
| Gender |  |  |
| Female | 2005 | 76.82 |
| Male | 605 | 23.18 |
| Professions |  |  |
| Doctor | 523 | 20.04 |
| Nurse | 1448 | 55.48 |
| Medical technician | 419 | 16.05 |
| Administration staff | 220 | 8.43 |
| Highest education attainment |  |  |
| Diploma or associate degree | 569 | 21.80 |
| Bachelor's degree | 1486 | 56.93 |
| Master's Degree | 375 | 14.37 |
| Doctorate | 180 | 6.90 |
| Income* |  |  |
| < = ¥200k | 1654 | 63.37 |
| ¥200 – 300k | 9 | 34.14 |
| ¥300 – 500k | 891 | 2.15 |
| > = ¥500k | 56 | .34 |
| Directly work with COVID-19 patients |  |  |
| Yes | 1296 | 49.66 |
| No | 1314 | 50.34 |
| Directly work with COVID-19 suspected patients |  |  |
| Yes | 1506 | 57.70 |
| No | 1104 | 42.30 |

*The average individual income in Beijing in 2021 is ¥75,002 RMB.

## Measures

Participants were instructed to complete an online questionnaire comprised of the Perceived Stress Scale, the Impact of Event Scale-Revised, the Reflective Functioning Questionnaire, and the Psychological Well-being Scale. All the measures used were Chinese versions. Demographic information of participants including age, gender, work, income, educational level, professions in hospitals, levels within professionals, and risks to be exposed to COVID-19 was also collected as part of the questionnaire. The measures can be viewed in S1 Appendix.

The Perceived Stress Scale (PSS) [83]. The PSS is a 14-item self-report scale measuring one's perception of pressure within 30 days. It is one of the most widely used scales to measure psychological stress. An item example is "In the last month, how often have you felt nervous and 'stressed'". Among the 14 items, seven items index tension subscale, and seven items index out of control subscale. Participants were instructed to report their feelings from 0 (never) to 4 (very frequently). The sum of the scores represents the perceived level of stress. A higher total score indicates a higher level of perceived stress. The psychometric properties of the measure have been extensively evaluated in various cultures and countries [89–92]. In this study, the Cronbach's alpha of the measure was .881.

The Impact of Event Scale-Revised (IES-R) [93]. The IES-R is a self-reported measure with 22 items to assess symptoms of posttraumatic stress disorder after traumatic event experience. It has been one of most widely used measures in the trauma literature and has been adapted to

measure traumatic stress symptoms related to COVID-19 [93–95]. In this study, the event in the questionnaire was specified as the participant's front-line-work experience in major outbreaks during the pandemic. The first item compared to IES-R was modified to "any reminder related to COVID-19 brought back feelings about it". Items are rated on a 5-point scale ranging from 0 (not at all) to 4 (extremely). An example item is "I was jumpy and easily startled". For coding and analysis, item scores can be summed up to yield a total score (ranging from 0 to 88) and three subscales of Intrusion, Avoidance, and Hyperarousal. In this study, only the total score was used. Higher scores indicate a higher level of the psychological impact of the situation concerning COVID-19. IES-R has been proven to have great reliability and validity [96]. The measure has been translated and well-validated in the Chinese population [5, 97, 98]. The Cronbach's alpha of the measure in this study was .97.

The Reflective Functioning Questionnaire (RFQ-8) [99]. The RFQ-8 is a self-report questionnaire for measuring mentalization. The measure includes 8 items that can be categorized into two subscales—RFQ certainty (RFQc) and RFQ uncertainty (RFQu). Each subscale has six shared items and two unique items. Participants were instructed to indicate their levels of agreement on statements using a 7-point Likert scale (1 = completely disagree, 7 = completely agree). The items were recoded to 3, 2, 1, 0, 0, 0, 0, to capture the extreme level of certainty. High scores of RFQc reflect hypermentalizing, while lower scores reflect a more genuine mentalizing. Similarly, items of RFQu are rated on a 7-point Likert scale and subsequently recoded to 0, 0, 0, 0, 1, 2, 3. Higher scores of RFQu indicate hypomentalizing whereas lower scores reflect an acknowledgment of the opaqueness of one's mental states and that of others, which is typical of genuine mentalizing. For instance, one item in the scale is "I don't always know why I do what I do". A strong rejection of the item is an indication of high certainty = 3 (i.e., hypermentalizing) on RFQc and at the same time indicative of low uncertainty = 0 on the uncertainty scale. A strong agreement of such statement would yield low certainty = 0 on RFQc and high uncertainty = 3 on RFQu. The measure has shown good reliability and validity in various cultural contexts [99–102]. The Cronbach's alpha in this sample was .880 for RFQc and .745 for RFQu respectively.

The Scale of Psychological Well-being (PWB) [103]. Ryff's Psychological Well-being Scale measures six aspects of well-being and happiness: autonomy, environmental mastery, personal growth, positive relations with others, the purpose of life, and self-acceptance. The PWB version used in this study is the shortened version with 18 items. An example of the items is "I like most parts of my personality". Participants rated their agreement with the statements on a 7-point Likert scale (1 = Strongly disagree, 7 = Strongly agree). This study used the sums to indicate the general level of psychological well-being. The 18-item version of the measure has good reliability and validity, and it has been widely used in different racial and ethnic groups [104, 105]. It has also been validated in the Chinese population [106, 107]. The Cronbach's alpha of PWB in this study was .901.

## Procedures

The present study has been approved by the Institutional Review Board of Beijing Anding Hospita (#2020–85). Posters of the study were first sent to the Human Resource Departments of all the participating hospitals. Each HR Department then informed all the relevant hospital departments about the study. Following which, the poster was circulated by administrative staff of each department to all the staff within their own social-media network (i.e. Wechat platform). Participants were invited to scan a QR code from the poster to complete the online survey. All the respondents who agreed to participate were informed about the purpose of the study and gave their written informed consent before completing the survey. Participation was

voluntary with no financial reward involved in completing the survey. All the items in the survey were required.

Due to the length of the survey in this project, the baseline data collection process was divided into two parts with one week apart between survey distributions. Once data were collected, the database was cleaned and coded before analysis. Participants who did not experience the major outbreak incidents in China, could not be matched with the first part of the survey respondents, those who did not complete the key questionnaires of the measures in this study, or who completed the questionnaires in less than 300 seconds or more than 3000 seconds were excluded from this study. A total of 3511 participants were recruited from 22 hospitals. After cleaning, 2610 participants met all the criteria for inclusion and their responses were included in this study.

## Data analysis

Descriptive statistics were used to summarize the characteristics of the sample to understand the demographics and central tendency of the key variables. Skewness and kurtosis of the all concerned variables were included to examine the distributions for symmetry and whether the data were heavy-tailed or light-tailed relative to a normal distribution. To explore the possible demographic differences on the measures of PSS, IES-R, RFQ-8 and PWB, t-tests were performed to examine the effects of gender on perceived stress, the impact of event, RFQ-8 (RFQc & RFQu), and psychological well-being. Levene's test for equality of variances were included in the t-tests to verify the assumption of equal variances across groups. One-way MANOVA tests were used to examine the effect of professions on the key variables. Post-hoc analyses were followed when significant differences were detected. To examine the relationships among the measures, Pearson's correlation was used. The serial mediation analyses of the hypothesized model were conducted using the PROCESS macro version 3.5 [108, 109] in SPSS. All the scores in the model analyses were standardized using z scores before running the mediation models. Separate analyses were conducted in the models using the two subscales of the RFQ. In each mediation model analysis, perceived stress was entered as the predictor, IES-R scores, RFQ (RFQc and RFQu) as mediators, and PWB as the main outcome in Model 6. Gender, age, exposure risk, professions, and levels within professionals were included in the analyses as covariates. In the process, 5000 bootstrap resamples of the effect size were used to determine the 95% confidence interval. All the analyses were conducted in the SPSS version 25.0.

## Results

### Comparison of mental health outcomes among the study sample

The t-test and ANOVA results were summarized in Tables 2 and 3. There was no significant effect of gender on perceived stress, RFQu, and psychological well-being. However, it was

**Table 2. Comparison of mental health outcomes between male and female HCWs (N = 2610).**

|  | Female (N = 2005) | | Male (N = 605) | | t | p | Cohen's d |
|---|---|---|---|---|---|---|---|
|  | M | SD | M | SD |  |  |  |
| 1. Perceived Stress | 36.83 | 7.28 | 36.37 | 7.89 | -1.33 | .183 | -.062 |
| 2. Impact of Event | 24.18 | 16.17 | 26.37 | 17.44 | 2.87 | .004* | .133 |
| 3. Reflective Functioning Certainty (RFQc) | 4.08 | 4.46 | 4.53 | 5.08 | 2.10 | .036* | .097 |
| 4. Reflective Functioning Uncertainty (RFQu) | 1.62 | 2.34 | 1.53 | 2.34 | -.807 | .420 | -.037 |
| 5. Psychological Well-being | 92.65 | 13.79 | 93.66 | 15.16 | 1.55 | .122 | .072 |

* significance level $p < .05$

**Table 3. Comparison of mental health outcomes across different professions.**

|  | Doctors (N = 523) | | Nurse (N = 1448) | | Medical technician (N = 419) | | Administration staff (N = 220) | | F | p | η2 |
|---|---|---|---|---|---|---|---|---|---|---|---|
|  | M | SD | M | SD | M | SD | M | SD |  |  |  |
| 1. Perceived Stress | 37.22 | 6.87 | 36.49 | 7.42 | 37.25 | 7.572 | 36.11 | 8.38 | 2.46 | .061 | .003 |
| 2. Impact of Event | 23.50 | 15.66 | 24.31 | 16.40 | 27.39 | 17.01 | 24.87 | 17.54 | 4.94 | .002* | .006 |
| 3. RFQc | 4.35 | 4.44 | 4.06 | 4.63 | 3.95 | 4.46 | 5.00 | 5.09 | 3.25 | .021* | .004 |
| 4. RFQu | 1.45 | 2.03 | 1.72 | 2.47 | 1.50 | 2.33 | 1.34 | 2.16 | 3.23 | .022* | .004 |
| 5. Psychological Well-being | 92.76 | 13.65 | 92.93 | 14.11 | 91.81 | 13.86 | 94.94 | 15.61 | 2.39 | .067 | .003 |

* significance level p < .05

significant on the impact of events and reflective functioning certainty. Female HCWs (M = 24.18, SD = 16.17) were reported to have significant lower mean scores of the impact of event than their male colleagues (M = 26.37, SD = 17.44), t (2608) = 2.87, p = .004, Cohen's d = .13. Similarly, females (M = 4.08, SD = 4.46) reported significantly lower scores of RFQc or more genuine mentalization than males (M = 4.53, SD = 5.08), t (2608) = 2.10, p = .036, Cohen's d = .097. In both cases, the effect sizes were very small. More detailed information and effect sizes were reported in Table 2.

Overall, the impact of event results showed that 51.1% of HCWs in this sample have a clinical concern for PTSD (IES-R> = 24 [110]; and 31.9% reached a probable diagnosis of PTSD (IES-R > = 33 [96]; The differences on the impact of event across different positions were significant, F (3, 2606) = 4.94, p = .002, η2 = .006. Subsequent Post-Hoc test results indicate that medical technicians (M = 27.39, SD = 17.01) reported a significantly higher level of IES than doctors (M = 23.50, SD = 15.66) and nurses (M = 24.31, SD = 16.40). There were also significant results found on RFQc (F (3, 2606) = 3.25, p = .021, η2 = .004) and RFQu (F (3, 2606) = 3.23, p = .022, η2 = .004) among different professions. Post Hoc tests showed that administration staff reported significantly higher scores of RFQc (M = 5.00, SD = 5.09) or hypermentalizing than nurses (M = 4.06, SD = 4.63) and medical technicians (M = 3.95, SD = 4.46). Nurses reported (M = 1.72, SD = 2.47) reported significantly higher scores of RFQu or hypomentalizing than doctors (M = 1.45, SD = 2.03) and administration staff (M = 1.34, SD = 2.16). The effect sizes reported on IES, RFQc and RFQu range from small to medium.

## Correlations among main variables

Pearson's correlation analyses of the main variables were summarized in Table 4. The results showed that perceived stress was positively associated with IES (r (2610) = .514, p < .001) and

**Table 4. Correlations of main variables.**

|  | M | SD | 1 | 2 | 3 | 4 |
|---|---|---|---|---|---|---|
| 1. Perceived Stress | 36.73 | 7.43 | — |  |  |  |
| 2. Impact of Event | 24.69 | 16.50 | .514** | — |  |  |
| 3. Certainty (RFQc) | 4.18 | 4.61 | -.488** | -.508** | — |  |
| 4. Uncertainty (RFQu) | 1.60 | 2.34 | .250** | .190** | -.222** | — |
| 5. Psychological Well-being | 92.88 | 14.12 | -.676** | -.457** | .508** | -.147** |

*significance level p < .05.

**significance level p < .01.

RFQu (r (2610) = .25, p < .001), whereas negatively associated with RFQc (r (2610) = -.488, p < .001) and psychological well-being (r (2610) = -.676, p < .001). Psychological well-being was positively related to RFQc (r (2610) = .508, p < .001), however negatively related to IES (r (2610) = -.457, p < .001) and RFQu (r (2610) = -.147, p < .001). IES was positively linked to RFQu (r (2610) = .190, p < .001), whereas negatively linked to RFQc (r (2610) = -.508, p < .001). Beyond that, RFQc and RFQu were negatively correlated to each other, r (2610) = -.222, p < .001.

### Mediation models and hypothesis testing

As described in the methods section, PROCESS was used to run the mediation models with gender, age, professions, and levels within professionals as covariates. RFQc and RFQu were tested separately as indicators for mentalization in the analyses. All the coefficients reported below were standardized coefficients. Symbol β refers to the regression coefficient between variables in the model and B refer to the indirect effect coefficient of perceived stress on psychological well-being.

The results of using RFQc as the indicator for mentalization in the model analysis were reported in Fig 2 and Table 5. The results showed that perceived stress had a significant and negative direct impact on the psychological well-being of the HCWs (β = -.54, SE = .02, t = -31.53, p < .001). The impact of event was found to be a significant mediator between perceived stress and psychological well-being (B = -.0412, SE = .0099, 95% CI [-.0611, -.0221]). In this model, perceived stress was positively associated with impact of event (β = .51, SE = .02, t = 30.68, p < .001), while impact of impact of event had a significant negative impact on

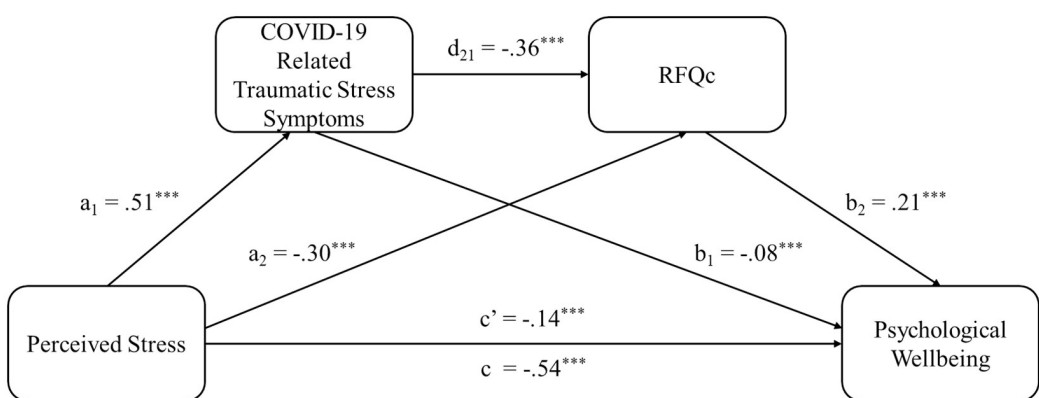

**Fig 2. Results of the serial mediation model showed the indirect effect of COVID-19-related traumatic stress symptoms, and the subsequent indirect effects of RFQc, in the relationship between perceived stress and psychological well-being.**

**Table 5. Results of the serial mediating effect with RFQc based on bootstrapping test (95% CI).**

|  | Effect | SE | LLCI | ULCI |
|---|---|---|---|---|
| Direct Perceived Stress Perceived Stress→Psychological Well-being | -.5351 | .0170 |  |  |
| Total indirect Perceived Stress Perceived Stress→Psychological Well-being | -.1415 | .0128 | -.1668 | -.1165 |
| Perceived Stress→Impact of event→Psychological Well-being | -.0412 | .0099 | -.0611 | -.0221 |
| Perceived Stress→RFQ Certainty→Psychological Well-being | -.0624 | .0075 | -.0777 | -.0488 |
| Perceived Stress→Impact of event→RFQ Certainty→Psychological Well-being | -.0380 | .0046 | -.0475 | -.0296 |

**Table 6. Results of the serial mediating effect with RFQu based on bootstrapping test (95% CI).**

|  | Effect | SE | LLCI | ULCI |
|---|---|---|---|---|
| Direct Perceived Stress Perceived Stress→Psychological Well-being | -.6049 | .0169 |  |  |
| Total indirect Perceived Stress Perceived Stress→Psychological Well-being | -.0717 | .0110 | -.0938 | -.0509 |
| Perceived Stress→Impact of event→Psychological Well-being | -.0808 | .0104 | -.1019 | -.0615 |
| Perceived Stress→RFQ Uncertainty→Psychological Well-being | .0074 | .0034 | .0014 | .0149 |
| Perceived Stress→Impact of event→RFQ Uncertainty→Psychological Well-being | .0017 | .0011 | .0001 | .0047 |

psychological well-being ($\beta$ = -.08, SE = .02, t = -4.62, p < .001). Therefore, perceived stress hindered the psychological well-being of HCWs by increasing the impact of event or traumatic symptoms related to COVID-19. In another mediation model, RFQc was found significantly mediating between the relationship between perceived stress and psychological well-being (B = -.0624, SE = .0075, 95% CI [-.0777, -.0488]). Perceived stress had a negative impact on RFQc ($\beta$ = -.30, SE = .02, t = -16.25, p < .001), whereas RFQc had a positive impact on psychological well-being ($\beta$ = .21, SE = .02, t = 12.12, p < .001). Furthermore, in the serial mediation analysis, the results showed that impact of event was closely linked to RFQc ($\beta$ = -.36, SE = .02, t = -19.18, p < .001). Perceived stress had a significant indirect effect on psychological well-being through impact of event and RFQc (B = -.0380, SE = .0046, 95% CI [-.0475, -.0296]). This serial mediation model of the two mediators on the relationship between perceived stress and psychological well-being explains 50.31% of the variance of psychological well-being, $F_{(7, 2602)}$ = 376.41, p < .001. The model indicates higher perceived stress predicts poorer psychological well-being through more severe traumatic symptoms, which in turn was linked with lower RFQc.

The results of using RFQu as an indicator for mentalization was also tested in mediation models (Table 6 and Fig 3). Similarly, to the previous models, perceived stress had a significant direct effect on psychological well-being of HCWs ($\beta$ = -.60, SE = .02, t = -35.85, p < .001), and impact of event significantly mediated between perceived stress and psychological well-being (B = -.0808, SE = .0104, 95% CI [-.1019, -.0615]). RFQu also independently mediated the association between perceived stress and psychological well-being (B = .0074, SE = .0034, 95% CI [.0014, .0149]). The higher perceived stress reduced psychological well-being by increased RFQ uncertainty. Between the two mediators, impact of event was closely linked to RFQu

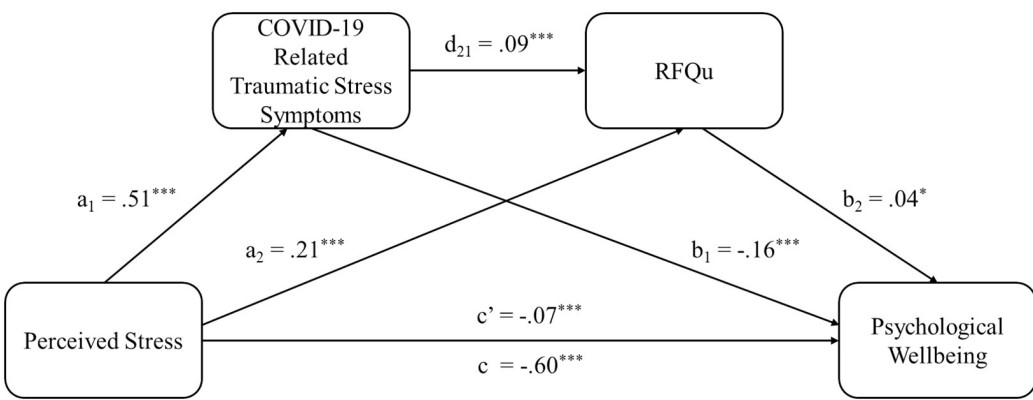

**Fig 3. Results of the serial mediation model showed the indirect effect of COVID-19-related traumatic stress symptoms, and the subsequent indirect effects of RFQu, in the relationship between perceived stress and psychological well-being.**

(β = .09, SE = .02, t = 4.03, p < .001). The serial mediating effect of the impact of event and RFQu was significant between perceived stress and psychological well-being (B = .0017, SE = .0011, 95% CI [.0001, .0047]. It was found that higher perceived stress inhibited psychological well-being through heightening impact of event, which in turn was linked with higher RFQu. The serial mediation model explained 47.64% of the variance of psychological well-being, F (7, 2602) = 338.0427, p < .001.

## Discussion

As mentioned previously, various factors including workplace environment, job-related stress [7, 9–14] individual's coping strategies or mechanism [15, 16], and social and organizational support [9, 17–22] can all affect the psychological well-being of HCWs. This study aimed in particular to investigate the mediating effects of COVID-19-related traumatic stress symptoms and mentalization on the relationship between perceived stress and psychological well-being in the population of health workers in China during the pandemic.

The PTSD prevalence of 31.9% in this study continues to highlight the long-lasting COVID-19 impact on the HCW population. It is consistent with the high prevalence of PTSD found in the meta-analysis among doctors (26%, 95% CI: 12% -40%) and nurses (36%, 95% CI: 25%–47%) in China [111]. The ongoing effect of COVID-19 demonstrated to have placed further strains on the mental health of the HCW population. The finding of significant gender effects on COVID-19-related traumatic stress symptoms was consistent with the literature that often sociodemographic variables such as gender were associated with psychological distress in HCWs during the pandemic [5]. However, more severe traumatic stress symptoms reported by male HCWs were different from previous studies in which female HCWs were found prone to experience psychological distress [112]. One explanation for the gender differences might be that during the pandemic, female HCWs were able to recover more quickly than their male colleagues. As evidenced in a longitudinal study of a lockdown in Spain, although women reported having worse psychological symptoms at the beginning, they improved significantly faster, and the gender differences were quickly diminished over the first few weeks [113]. In terms of job responsibilities, it was found in this study that medical technicians reported a significantly more severe traumatic stress symptoms than doctors and nurses. One possible reason might be compared to nurses and doctors, medical technicians receive more limited training but are similarly confronted with a high risk of exposure and heavy workload. As found in a study in Canada, medical laboratory professionals experienced high rates of burnout during the pandemic [114]. The finding in our study therefore is important to draw attention to non-clinical staff including medical technicians and administrative staff as they were often overlooked in studies involving HCWs and it contrasts with previous findings that front-line medical staff were more likely to experience psychological distress than non-clinical staff [115].

### Mentalization

The brief RFQ-8 measure captures the spectrum of impaired mentalizing capacity including hypo- and hypermentalizing. In this study, the negative correlation between RFQu and RFQc supports the measures' proposed opposition structure of too little or too much certainty about one's interpretation of mental states of self and others as well as the conceptualization of mentalization with two maladaptive poles of hypo- and hypermentalizing. In the results, hypomentalizing was found positively associated with perceived stress and traumatic stress symptoms however, negatively associated with psychological well-being. In other words, for HCWs during the pandemic, the inability to mentalize was more likely to be linked to higher perceived

stress and more traumatic stress symptoms, and poorer psychological well-being, whereas genuine mentalizing was associated with less perceived stress, less severe traumatic stress symptoms and better psychological well-being. This finding supports the literature and most of the past studies on mentalization that hypomentalizing have shown to increase the vulnerability of psychopathology [71, 99], and strong and genuine mentalizing abilities were found to act as a protective for mental health [59, 68], related to resilience [61] and psychological well-being [67]. In particular, it is consistent with the finding of a study in Serbia that hypomentalizing explained part of the variance of burnout dimensions of emotional exhaustion and depersonalization in HCWs and it was negatively related to personal accomplishment [57]. In a different population of pre-pandemic entrepreneurs, it was also found that hypomentalizaing attributed to cynicism and emotional exhaustion and negatively associated with professional efficacy [86]. Genuine mentalizing capacity allows HCWs communicate and interact with patients and colleagues with empathy and sincere curiosity about self and others' mental states which helps them to overcome stress [68, 85]. Hypomentalizing refers to lack or absence of the social meaning making process that can result in wrong conclusions or understanding of self and other's mental states and behaviors which can subsequently interfere with their relationship with families, friends, colleagues and patients.

Hypermentalizing or too much mentalizing were found unexpectedly in this study negatively associated with perceived stress, traumatic stress symptoms and positively associated with psychological well-being. This finding does not support the previous assumption that hypermentalizing was associated with increased vulnerability to psychopathology [71, 99]. However, similar results were found in Safiye and colleagues' study [76] that hypermentalization was unexpectedly negatively associated with emotional exhaustion and depersonalization, and positively associated with personal accomplishment in HCWs during the COVID-19 pandemic. Hypermentalizing involves building extensive theories about their own and others' mental states without the basis on facts [116]. The finding may be understood as that this tendency to overly make assumptions beyond objective information or over-interpretation of the mental states of others is not a characteristic of HCWs. Especially the nature of HCWs' work requires them to be objective and show confidence in their work and judgement. Another explanation of such unexpected results might be a question about the validity of the RFQ-8 construct. Over the years even though RFQ-8 was used widely in the field, the validity of the construct was still criticized for its scoring procedure and two-pole dimensionality [117, 118]. In particular, Müller and colleagues in their study and review pointed out that existing evidence of the measurement validity is largely based on the general pattern of association that RFQu is related to poorer mental health and found in their empirical test the RFQ-8 measure may only assess as single latent dimension related to hypomentalizing and unlikely to capture maladaptive forms of hypermentalizing [117]. It might be possible that hypermentalizing was not captured accurately by RFQ-8 here in this HCW population or RFQc was assessing something else that is closer to certainty or confidence of their ability to make judgement rather than the excessive certainty in accuracy of one's own beliefs about the nature of mental states that underlie one's behavior [99].

## The mediators of COVID 19-related traumatic stress symptoms and mentalization

In the mediation models, perceived stress during COVID-19 had a direct strong negative effect on the psychological well-being of HCWs as predicted. This is consistent with the literature across cultures that perceived stress has a significant impact on the psychological well-being of HCWs [4, 56, 119] and higher level of perceived stress is associated with psychological distress

[54, 55]. Regarding the second hypothesis, COVID-19 related traumatic stress symptoms was found as a significant mediator in the relationship between perceived stress and psychological well-being. Specifically, traumatic stress increased the strength of the negative effect that perceived stress had on psychological well-being. Taken the robust connection between perceived stress and psychological well-being of HCWs together with the evidence of the strong relationship between perceived stress and traumatic stress symptoms of this population during the pandemic [120, 121], the mediation model helps to clarify the role of the traumatic stress symptoms caused by the specific working experience in the healthcare setting of the COVID-19 pandemic. As perceived stress can refer to more generalized source of stress, it is one of the psychological susceptibility factors that can interact or mingle with current traumatic stress of the pandemic and subsequently affect the psychological well-being of HCWs. In other words, HCWs who had traumatic experience fighting COVID-19 in the past were more likely to have negative psychological well-being when facing with the higher levels of psychological susceptibility.

The third hypothesis was in partial as predicted. In the mentalization mediation model, both hypomentalizing and hypermentalizing were found as significant mediators. However, higher perceived stress hindered psychological well-being of HCWs by increased hypomentalizing; on the other hand, higher perceived stress was more likely to be associated with worse psychological well-being through less hypermentalizing. It is understandable that when perceived stress is higher, HCWs may have reduced or even lose the ability to mentalize self and others' mental states which can lead to more burnout including emotional exhaustion, depersonalization, cynicism and less sense of personal accomplishment and professional efficacy [76, 86], which in turn can cause severe impairment on their psychological well-being. More genuine mentalizing, on the other hand, can help overcome stress [68, 85] and reduce burnout [76, 86]. It is more puzzling to understand the role of hypermentalizing in this study. As discussed earlier regarding the validity of RFQ-8 measure on the hypermentalizing construct, it might be possible that hypermentalizing captured in this study is closer to adaptive mentalizing. It should also be noted that the coefficient in the mediation model analyses between hypomentalizing and psychological well-being appeared positive as opposed to the negative correlation coefficient between hypomentalizing and psychological well-being. This is because the mediation models used standardized z scores of all measures whereas the correlations used raw scores. As the hypomentalizing variable is introduced in the models after other variables in sequence, the coefficient had also changed the direction between hypomentalizing psychological well-being.

Finally, concerning the fourth hypothesis, there was a partial serial mediation of COVID-19-related traumatic stress symptoms and mentalization on the relationship between perceived stress and the psychological well-being of HCWs. Specifically, higher perceived stress negatively affected psychological well-being of HCWs through more severe traumatic stress symptoms related to COVID-19, which in turn linked with more hypomentalizing or less hypermentalizing. For hypomentalizing model, the results showed that HCWs who had more severe traumatic experience working at the frontline fighting COVID-19, may be more likely to suffer poorer psychological outcomes as they lose their mentalizing capacity while facing daily stress. As previous theory has pointed out, mentalizing ability can decrease as stress and arousal increase [78]. In stressful situations such as dealing with critical incidents during the COVID-19 pandemic, HCWs may regress to a pre-mentalizing model and mentalizing deficits become more pronounced as the arousal level rises [61]. It is also consistent with the theory and clinical observation in the context of COVID-19 that even people with stable and high mentalizing capacity may even lose their mentalizing ability due to stress [80]. It should be also noted that people impacted by trauma are more likely to experience severe arousal, which

can also cause a loss of mentalization [81, 82]. The finding further supports that genuine mentalizing (low hypomentalizing) can provide valuable psychological resources to promote psychological well-being [67] and act as an effective protective factor for psychopathologies especially for individuals exposed to trauma [59, 69]. For hypermentalizing, one tentative hypothesis might be that for HCWs facing tremendous stress every day in a time of crisis, excessive mentalizing might be a more adaptive mentalizing stance, especially for HCWs who continue to work on the frontline despite suffering from PTSD. Therefore, as demonstrated in the model, it acted more closely to the effect of genuine mentalizing ability.

## Implications

As most of the literature has focused on the impact of the pandemic on psychological distress in healthcare settings [2–6] as well as the role of perceived stress in relation to various psychological problems experienced by HCWs [21, 122], there has not been any study investigating the relationship between the perceived stress and psychological well-being with COVID-19 related traumatic stress symptoms and mentalizing capacity. The current study contributes to the literature of how the COVID-19 pandemic influences the positive functioning of HCWs in life including autonomy, personal growth, positive relationships, environmental mastery, purpose in life and self-acceptance [41]. The most important theoretical implication of the findings here is regarding mentalization and its forms of hypomentalizing and hypermentalizing. Mentalization is often found to be significantly influenced by the early traumatic experience [57, 63], the findings from this study suggested a changeable nature of mentalization, which fluctuates over time and across different contexts [66]. It responds to the stress and arousal in the environment [61, 79]. The current study extended the existing literature by confirming the impact of trauma on mentalization beyond the attachment relationship [71] as well as providing evidence for the connection between trauma and mentalization beyond the clinical population. We recognized the effect of trauma on mentalization from later life experiences, specifically, the COVID-19 pandemic as a collective trauma for HCWs in the present study. This supports the theory and clinical observation that the increased stress caused by the COVID-19 pandemic can lead to the temporary loss of mentalizing ability [80].

Another approach to look at hypermentalizing and hypomentalizing in this study is perhaps these two forms captured by RFQ-8 can be understood as HCWs'coping response to high levels of perceived stress in a collective trauma experience. Traumatic events or a critical incident cause individuals to experience unbearable emotional responses for which usual coping mechanisms are ineffective [27]. HCWs who suffered COVID-19-related traumatic stress symptoms may be more susceptible to high perceived stress and cope with the stress by decreasing their mentalizing capacity to reduce negative emotions and conflicts. Although there is no existing literature connecting mentalizing with coping, there has been evidence that individuals may use coping in relation to their appraisals of stressful events and personal resources, which subsequently influence well-being measures [123]. This is in line with previous findings that cognitive (problem-focused) and motivational (meaning-focused) coping strategies can effectively mediate the relationship between various risk perception and psychological well-being [124, 125], whereas emotional coping proved to be a less significant way of coping with stress during the pandemic [124]. Therefore, although hypomentalizing eventually may lead to impaired psychological well-being including emotional exhaustion, depersonalization and disruption of interpersonal relationships [76], it might be an important coping strategy for HCWs reacting to the stress during the pandemic. Hypomentalizing may be a temporary strategy they need to continue working daily in a healthcare setting after being exposed to traumatic events even if it is at the expense of their long-term psychological well-

being. As previously discussed, hypermentalizing construct measured in this study might be a more adaptive coping stance for HCWs in the current context. For HCWs who have to be vigilant every day assessing potential risks of COVID-19 infection on self and others, the certainty or excessive mentalizing of self and others' mental states might be effective for the appraisal. In addition, it is also interesting in the finding that nurses tend to have significantly higher level of hypomentalizing whereas administration staff tend to have significantly higher level of hypermentalizing. This tendency may well reflect the nature of their work and which mentalizing stance can best carry out their job responsibilities.

In terms of the practical implication of study, the traumatic stress during the pandemic on HCWs has again been identified and calls for attention [4, 5, 15]. This study highlighted the importance of recognizing the pandemic as a collective trauma associated with perceived stress that could influence the mentalizing capacity and psychological well-being of clinical and non-clinical staff in the healthcare field. We emphasize that it is of great value and significance to focus on interventions that can facilitate the mentalizing capacity in the intervention in the current context. The ineffective use of hypomentalizing will result in HCW burnout and negative psychological well-being, thus, effective interventions for genuine mentalizing abilities could significantly help to cope with its failure in the extreme arousal context and further promote psychological functioning among the healthcare population and beyond [58, 59, 66, 87, 88]. We also believe as Lassri and Desatnik had advocated in their paper that "mentalization can offer a comprehensive integrative framework for a potentially widely implementing psychotherapy interventions for COVID-19-related psychopathology; and that interventions encompassing a mentalizing perspective will play an important role in mending societal and mental health wounds even after the COVID-19 crisis is over"[80]. In addition, as hypomentalizing impairs the positive psychological functioning of HCWs, more research or observations should be conducted to examine how hypomentalizing is expressed with patients and colleagues in the interpersonal context and relevant interpersonal skill training program can address some of the issues occurred [58]. Finally, it is important to acknowledge and recognize the shared trauma experience during the pandemic, any intervention and prevention support for HCWs should use a trauma-informed approach. Implementation of the mentalization framework and trauma-informed practice in the intervention work with HCWs will likely improve their ability to self-care and to nurture others. This is particularly crucial for the HCW population during and even beyond the pandemic.

## Limitations and future directions

The findings of the present study should be understood within several limitations. First, the cross-sectional design may not reflect the long-term change or causal links in perceived stress and psychological well-being as well the effects of the two mediators. Longitudinal research could be promising for future studies to clarify the relationship between perceived stress and psychological well-being in the changeable context of pandemic trends, as well as the mediating role of the impact of event and mentalization. Second, regarding the data collection process, all the items in the survey were required thus participants could not skip any question. Although such approach prevented missing values, it had ramifications on the authenticity of participant responses when they were not certain or even uncomfortable to answer questions. It had also impacted the completion rate of the responses. In future studies, some items in the survey should be made optional. Third, the key variables in this study were measured by self-report scales to capture the subjective experience of participants. Common-method bias was tested in this study using Herman's one-factor test. Of all the factors in the study, it was found the total variance extracted by one factor is 31.943% (<50%) which shows common-method

bias is not present in this study. However, future studies could consider extending the current finding by using more objective or alternative measures such as peer-report, physical indexes, organizational indexes, or different versions of questionnaires with more validity. Forth, there are limitations in terms of specific measures used in the study. The RFQ-8 used to measure mentalizing in this study was also found with questions, especially in the domain of certainty (i.e., hypermentalizing). Although this measure has been used widely across different culture contexts [99–102] and it showed good internal consistency in this study, it has not been well-validated in the Chinese culture. While the results in this study need to be interpreted with that limitation in mind, future studies will be required to establish the validity and reliability of this measure as well as to better understand the mentalization mechanisms of hypo- or hyper-mentalizing in the Chinese population. Subsequently, more culture relevant theory or inter-pretation could be used to understand the mediation models identified in this study. In terms of psychological well-being, this study used the total score of Ryff's PWB scale. The six dimen-sions of the construct were not examined separately. As hypomentalizing and hypermentaliz-ing had different relationships with psychological well-being, more detailed analysis will be beneficial to disentangle and understand the complex relationship between each dimension of psychological well-being and the two constructs of mentalization. Lastly, the findings in this study are limited to the specialized population of HCWs in the context of the COVID-19 pan-demic. Future studies with HCWs as well as a more general population can help to test whether such mediation models can be replicated. It will contribute to a wider application of mentalization informed preventions or interventions for promoting psychological well-being during and beyond the current crisis.

## Conclusions

Healthcare workers serve a vital role in public health, particularly during health care crises such as the current COVID-19 pandemic. There is an urgent need to understand the factors that can affect their immediate and long-term psychological well-being during and beyond the current pandemic. The present study focused on delineating the mechanisms underlying the relationship between perceived stress and psychological well-being in the healthcare workforce during the COVID-19 pandemic in China. The findings in this study help to identify impor-tant components of traumatic stress symptoms related to COVID-19 and mentalization as well as their roles in mediating perceived stress and psychological well-being. Based on our find-ings, we strongly recommend implementing trauma-informed practice with a mentalization framework in intervention work with HCWs as it will likely enhance their capacity to care for themselves and others. It will also help to prepare the healthcare field for other similar out-breaks in the future by better protecting the workforce with less traumatic impact and increased capacity to mentalize.

## Supporting information

**S1 Appendix. Appendix of measures.**
(DOCX)

**S1 Data.**
(CSV)

## Acknowledgments

We thank all the healthcare workers for their amazing work combatting the COVID-19 Pan-demic and their participation in this study.

## Author Contributions

**Conceptualization:** Qian Wang, You Zhou, Xinyu Pan.

**Investigation:** Zhe Wang, Yinqi Liu, Tengfei Tian, Sixiang Liang.

**Methodology:** Qian Wang, You Zhou.

**Validation:** Sha Sha.

**Writing – original draft:** Qian Wang, You Zhou, Xinyu Pan.

**Writing – review & editing:** Qian Wang, You Zhou, Gang Wang.

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
