## [Decision Letter · Decision Letter 0]

11 Jan 2024

PONE-D-23-31204The mediation effects of COVID-19-related traumatic stress symptoms and mentalization on the relationship between perceived stress and psychological wellbeing in healthcare workers transitioning to a post-pandemic worldPLOS ONE

Dear Dr. Wang,

Thank you for submitting your manuscript to PLOS ONE. After careful consideration, we feel that it has merit but does not fully meet PLOS ONE’s publication criteria as it currently stands. Therefore, we invite you to submit a revised version of the manuscript that addresses the points raised during the review process.

**ACADEMIC EDITOR: **

The authors have submitted a good manuscript. However, as noted by reviewers some edits are recommended as notes: "My only critique would be that the paper's focus on collective trauma without considering other stressors impacting healthcare workers' psychological wellbeing. It is critical to evaluate a range of factors, such as individual coping mechanisms, social support, and workplace environment, to provide a holistic understanding of the wellbeing of healthcare workers."If these revisions are incorporated the manuscript can be considered for publication

We look forward to receiving your revised manuscript.

Kind regards,

Souparno Mitra, M.D.

Academic Editor

PLOS ONE

2. We noticed you have some minor occurrence of overlapping text with the following previous publication(s)(but not limited to these, which needs to be addressed:

a) Lassri, D. and Desatnik, A., 2020. Losing and regaining reflective functioning in the times of COVID-19: Clinical risks and opportunities from a mentalizing approach. Psychological Trauma: Theory, Research, Practice, and Policy, 12(S1), p.S38.

b) Zhou, T., Xu, C., Wang, C. et al. Burnout and well-being of healthcare workers in the post-pandemic period of COVID-19: a perspective from the job demands-resources model. BMC Health Serv Res 22, 284 (2022). https://doi.org/10.1186/s12913-022-07608-z

In your revision ensure you cite all your sources (including your own works), and quote or rephrase any duplicated text outside the methods section. Further consideration is dependent on these concerns being addressed.

Reviewers' comments:

Reviewer's Responses to Questions

**Comments to the Author**

1. Is the manuscript technically sound, and do the data support the conclusions?

Reviewer #1: Yes

Reviewer #2: Yes

2. Has the statistical analysis been performed appropriately and rigorously? 

Reviewer #1: Yes

Reviewer #2: Yes

3. Have the authors made all data underlying the findings in their manuscript fully available?

Reviewer #1: Yes

Reviewer #2: Yes

4. Is the manuscript presented in an intelligible fashion and written in standard English?

Reviewer #1: Yes

Reviewer #2: Yes

5. Review Comments to the Author

Reviewer #1: The research paper under discussion aims to explore the effects of perceived stress and COVID-19-related traumatic stress symptoms on healthcare workers' psychological wellbeing, and how mentalization might mediate this relationship. The paper defines key concepts such as perceived stress, traumatic stress symptoms, and mentalization, which aids in understanding their role in the study. It is a well written paper. My only critique would be that the paper's focus on collective trauma without considering other stressors impacting healthcare workers' psychological wellbeing. It is critical to evaluate a range of factors, such as individual coping mechanisms, social support, and workplace environment, to provide a holistic understanding of the wellbeing of healthcare workers.

Reviewer #2: Excellent manuscript with clear explanation of context, scales and results. I would think that post traumatic symptoms and mentalization would mediate the effects on psychological well being of perceived stress but it is good to see it being demonstrated in your results.

6. PLOS authors have the option to publish the peer review history of their article (what does this mean?). If published, this will include your full peer review and any attached files.

Reviewer #1: No

Reviewer #2: No

---

## [Author Response · Author response to Decision Letter 0]

28 Jun 2024

That you for all the comments and reviews. We have also uploaded a word document with all the responses.

Responses to reviewers

Reviewer #1: The research paper under discussion aims to explore the effects of perceived stress and COVID-19-related traumatic stress symptoms on healthcare workers' psychological wellbeing, and how mentalization might mediate this relationship. The paper defines key concepts such as perceived stress, traumatic stress symptoms, and mentalization, which aids in understanding their role in the study. It is a well written paper. My only critique would be that the paper's focus on collective trauma without considering other stressors impacting healthcare workers' psychological wellbeing. It is critical to evaluate a range of factors, such as individual coping mechanisms, social support, and workplace environment, to provide a holistic understanding of the wellbeing of healthcare workers.

Author response: Thank you for this excellent suggestion. We have reframed and added content at the beginning of the introduction (line 62-140) before introducing collective trauma and other key concepts. Various factors were also emphasized in the discussion (line 695-698).We think the revision should be able to provide a more holistic view of understanding the psychological well-being of HCWs before focusing on collective trauma, perceived stress and mentalization. Thank you again for taking the time to review our manuscript.

Reviewer #2: Excellent manuscript with clear explanation of context, scales and results. I would think that post traumatic symptoms and mentalization would mediate the effects on psychological well being of perceived stress but it is good to see it being demonstrated in your results.

Author response: Thank you so much for your time to review our manuscript.

ACADEMIC EDITOR:

The authors have submitted a good manuscript. However, as noted by reviewers some edits are recommended as notes: "My only critique would be that the paper's focus on collective trauma without considering other stressors impacting healthcare workers' psychological wellbeing. It is critical to evaluate a range of factors, such as individual coping mechanisms, social support, and workplace environment, to provide a holistic understanding of the wellbeing of healthcare workers." If these revisions are incorporated the manuscript can be considered for publication

Author response: please see response to reviewer 1. Changes have been made in introduction and discussion of the manuscript.

Author response: we have reformatted the manuscript according to the templates and style guideline provided. Tables were added and fonts were adjusted in the main text.

2. We noticed you have some minor occurrence of overlapping text with the following previous publication(s)(but not limited to these, which needs to be addressed:

a) Lassri, D. and Desatnik, A., 2020. Losing and regaining reflective functioning in the times of COVID-19: Clinical risks and opportunities from a mentalizing approach. Psychological Trauma: Theory, Research, Practice, and Policy, 12(S1), p.S38.

b) Zhou, T., Xu, C., Wang, C. et al. Burnout and well-being of healthcare workers in the post-pandemic period of COVID-19: a perspective from the job demands-resources model. BMC Health Serv Res 22, 284 (2022).https://doi.org/10.1186/s12913-022-07608-z

In your revision ensure you cite all your sources (including your own works), and quote or rephrase any duplicated text outside the methods section. Further consideration is dependent on these concerns being addressed.

Author response: both references mentioned have been fixed in the manuscript, direct quotes were added for reference (a), reference (b) has also been added. All the references were checked and reformatted according to the journal requirement.

3. PLOS requires an ORCID iD for the corresponding author in Editorial Manager on papers submitted after December 6th, 2016. Please ensure that you have an ORCID iD and that it is validated in Editorial Manager. To do this, go to ‘Update my Information’ (in the upper left-hand corner of the main menu), and click on the Fetch/Validate link next to the ORCID field.

This will take you to the ORCID site and allow you to create a new iD or authenticate a pre-existing iD in Editorial Manager.

Author response: ORCID has been added in the profile.

Author response: this has been fixed. 

Author response: this has been fixed in the manuscript and figures uploaded. Captions have been added, thank you for noticing the missing captions.

6. Please include captions for your Supporting Information files at the end of your manuscript, and update any in-text citations to match accordingly.

Author response: S1 Appendix was added to the main manuscript.

Author response: All the references were checked and reformatted according to the journal requirement.

---

## [Decision Letter · Decision Letter 1]

14 Aug 2024

The mediation effects of COVID-19-related traumatic stress symptoms and mentalization on the relationship between perceived stress and psychological well-being in healthcare workers transitioning to a post-pandemic world

PONE-D-23-31204R1

Dear Dr. Wang,

We’re pleased to inform you that your manuscript has been judged scientifically suitable for publication and will be formally accepted for publication once it meets all outstanding technical requirements.

Kind regards,

Souparno Mitra, M.D.

Academic Editor

PLOS ONE

Additional Editor Comments (optional):

Reviewers' comments:

Reviewer's Responses to Questions

**Comments to the Author**

1. If the authors have adequately addressed your comments raised in a previous round of review and you feel that this manuscript is now acceptable for publication, you may indicate that here to bypass the “Comments to the Author” section, enter your conflict of interest statement in the “Confidential to Editor” section, and submit your "Accept" recommendation.

Reviewer #2: All comments have been addressed

Reviewer #3: All comments have been addressed

2. Is the manuscript technically sound, and do the data support the conclusions?

Reviewer #2: Yes

Reviewer #3: Yes

3. Has the statistical analysis been performed appropriately and rigorously? 

Reviewer #2: Yes

Reviewer #3: Yes

4. Have the authors made all data underlying the findings in their manuscript fully available?

Reviewer #2: Yes

Reviewer #3: Yes

5. Is the manuscript presented in an intelligible fashion and written in standard English?

Reviewer #2: Yes

Reviewer #3: Yes

6. Review Comments to the Author

Reviewer #2: (No Response)

Reviewer #3: Your study offers important insights into the psychological well-being of healthcare workers (HCWs) in China during the COVID-19 pandemic. It specifically focuses on the mediating effects of COVID-19-related traumatic stress symptoms and mentalization on the relationship between perceived stress and psychological well-being.

7. PLOS authors have the option to publish the peer review history of their article (what does this mean?). If published, this will include your full peer review and any attached files.

Reviewer #2: No

Reviewer #3: No

---

## [Editor Report · Acceptance letter]

22 Aug 2024

PONE-D-23-31204R1 

PLOS ONE

Dear Dr. Wang, 

I'm pleased to inform you that your manuscript has been deemed suitable for publication in PLOS ONE. Congratulations! Your manuscript is now being handed over to our production team.

Kind regards, 

on behalf of

Dr. Souparno Mitra 

Academic Editor

PLOS ONE